# Inhibition of β-lactamase function by *de novo* designed peptide

**Arunima Mishra** [1]*, **Irena Cosic**[2], **Ivan Loncarevic**[3], **Drasko Cosic**[2], **Hansel M. Fletcher**[1]*

**1** Division of Microbiology & Molecular Genetics, School of Medicine, Loma Linda University, Loma Linda, California, United States America, **2** AMALNA Consulting, Black Rock, Melbourne, VIC, Australia, **3** QuantBioRes-QBR A/S, Copenhagen, Denmark

\* amishra@llu.edu (AM); hfletcher@llu.edu (HMF)

**Data Availability Statement:** All data generated or analyzed during this study are included in the published article. The detailed sequences of peptides pep1-4 are not included in the article due

## Abstract

Antimicrobial resistance is a great public health concern that is now described as a "silent pandemic". The global burden of antimicrobial resistance requires new antibacterial treatments, especially for the most challenging multidrug-resistant bacteria. There are various mechanisms by which bacteria develop antimicrobial resistance including expression of β-lactamase enzymes, overexpression of efflux pumps, reduced cell permeability through downregulation of porins required for β-lactam entry, or modifications in penicillin-binding proteins. Inactivation of the β-lactam antibiotics by β-lactamase enzymes is the most common mechanism of bacterial resistance to these agents. Although several effective small-molecule inhibitors of β-lactamases such as clavulanic acid and avibactam are clinically available, they act only on selected class A, C, and some class D enzymes. Currently, none of the clinically approved inhibitors can effectively inhibit Class B metallo-β-lactamases. Additionally, there is increased resistance to these inhibitors reported in several bacteria. The objective of this study is to use the Resonant Recognition Model (RRM), as a novel strategy to inhibit/modulate specific antimicrobial resistance targets. The RRM is a bio-physical approach that analyzes the distribution of energies of free electrons and posits that there is a significant correlation between the spectra of this energy distribution and related protein biological activity. In this study, we have used the RRM concept to evaluate the structure-function properties of a group of 22 β-lactamase proteins and designed 30-mer peptides with the desired RRM spectral periodicities (frequencies) to function as β-lactamase inhibitors. In contrast to the controls, our results indicate 100% inhibition of the class A β-lactamases from *Escherichia coli* and *Enterobacter cloacae*. Taken together, the RRM model can likely be utilized as a promising approach to design β-lactamase inhibitors for any specific class. This may open a new direction to combat antimicrobial resistance.

## Introduction

The discovery of penicillin by Alexander Fleming in 1929 is one of the major biomedical breakthroughs in human history [1]. Since then, antibiotics have been recognized as a powerful drug and have directly saved numerous lives by enabling the treatment of once-common

to unresolved intellectual property challenges. The authors are willing to share the sequences in response to individual requests to Hansel Fletcher [hfletcher@llu.edu] and/or Ivan Loncarevic [ivlon@ymail.com].

**Funding:** RRM analysis, peptide design and peptide synthesis were financed by QuantBioRes-QBR A/S. Work on testing of peptides was supported by Public Health Services Grants DE030411 and DE025852 from NIDCR (to HMF) and DE029825 (to AM). The funder NIDCR had no role in study design, data collection and analysis, decision to publish, or preparation of the manuscript.

**Competing interests:** The authors have declared that no competing interests exist.

causes of death such as pneumonia and sepsis. In addition, its use has had a significant positive impact on a range of healthcare interventions such as surgery, chemotherapy, and organ transplants [2,3]. β-lactam antibiotics are the most often used antimicrobial agents and continue to play a central role in treating bacterial infections [4,5]. These drugs have a highly reactive β-lactam ring in their structure. At present, penicillins, cephalosporins, carbapenems and monobactams are the four main classes of β-lactam antibiotics in clinical use. They cause cell death by interrupting bacterial cell-wall formation by binding to essential penicillin-binding proteins, enzymes that are involved in the terminal steps of peptidoglycan cross-linking, in both Gram-positive and Gram-negative bacteria [6,7].

Like other antimicrobial classes, the widespread and excess use of β-lactams in clinical practice have led to antibiotic resistance in bacteria with a resultant big burden and extra cost on health-delivery systems [8]. In the last 15 years, the problem of antibiotic resistance to two or more drugs (multidrug-resistance) has increased exponentially, thus, challenging the management of severe healthcare-associated infections, increasing morbidity and mortality, and generating strains with extreme resistance [9,10]. More than 2.8 million antimicrobial-resistant infections, linked to nearly 35,000 deaths at a healthcare cost of approximately 2 billion dollars, have been recently reported in the United States of America [11]. The World Health Organization (WHO) has predicted that by 2050 deaths associated with multidrug-resistant bacteria will be 10 million people a year which is greater than the current global deaths due to all cancers (8.2 million) costing up to $100 trillion [12].

The antimicrobial resistance mechanisms involve both enzymatic and non-enzymatic reactions. The enzymatic mechanism includes the expression of enzymes which can inactivate the antibiotic. Non-enzymatic mechanisms may result from non-transmissible mechanisms (disabling the drugs, overexpression of efflux pumps, reduced cell permeability through downregulation of porins required for β-lactam entry or modifications in penicillin-binding proteins) or may be transmissible via transfer of mobile genetic elements such as plasmid-borne β-lactamases [6,13–15]. The most specific resistance mechanism among these is the production of β-lactamases by both Gram-positive and Gram-negative bacteria which hydrolyze the amide bond in the β-lactam ring and thus making the antibiotic ineffective [16]. On the basis of specific sequence motifs and hydrolytic mechanism, the Ambler Classification System has grouped β-lactamase enzymes into four classes named A, B, C, and D [17]. Classes A, C and D have serine at their catalytic sites (seine β-lactamases), while class B enzymes need $Zn^{2+}$ as a cofactor for their activity and are specifically called metallo-β-lactamases (MBLs). The major strategies to tackle β-lactamase mediated resistance are developing new antibiotics or improvements to existing β-lactams themselves and the use of combinations of susceptible β-lactams with β-lactamase inhibitors [18]. These inhibitors protect the β-lactam from β-lactamase hydrolysis thus restoring its antimicrobial potential.

Following the introduction of clavulanic acid [19] as the first β-lactamase inhibitor, penicillin-inhibitor combinations (amoxicillin-clavulanate, ampicillin-sulbactam, piperacillin-tazobactam) have been extensively used as treatments for infections caused by β-lactamase-producing bacteria [20]. While successful in increasing the potency of the β-lactams, the inhibitor combinations are only effective against selected class A enzymes such as TEM, SHV and the CTX-M classes [21–23]. Another newly approved combination such as ceftazidime-avibactam inhibits classes A, C and some of class D enzymes [6,7,18,24]. Class B MBLs pose a particular challenge because so far, none of the clinically approved inhibitors can effectively inhibit the members of this class [25–27]. The limitations of these current inhibitors warrant further research in an effort to develop more effective and likely broad-scope β-lactamase inhibitors against all the classes of β-lactamase enzymes.

Here, we have applied the Resonant Recognition Model (RRM) to design β-lactamase inhibitors. The RRM model is a powerful technique [28–31] that is based on the finding that there is a significant correlation between spectra of free electron energy distribution along protein and its biological activity. It has been previously shown that proteins with the same biological function or interactive activity have the same periodic components in the free electron energy distribution along with the protein molecule. Furthermore, it was found that the RRM frequencies represent the characteristic features of proteins' biological functions or interactions and thus they are relevant parameters for mutual recognition between biomolecules [28–31]. Therefore, the RRM frequencies are significant in describing the selectivity of interaction between proteins and their substrates or targets but are not describing chemical binding [28–31]. The RRM approach can be used to design β-lactamase inhibitor peptides for both broad spectrum enzymes as well as against a particular class due to the advantage that it does not look into structural characteristics of the binding domain but uses the full protein's biophysical parameters that are important for its binding activity. This approach has been successfully used to design peptides with desired biological function/interaction. These designs were experimentally tested in a number of applications [32–38] including the design of a peptide to mimic myxoma virus oncolytic function [32,35], as well as a recently developed peptide that prevents entry of the SARS-CoV-2 virus into the host cells [34,37].

In this study, we have utilized the RRM approach to design 30-mer bioactive peptides which can modulate β-lactamase activity. The *de novo* designed peptides pep3 (inhibitor) and pep1/pep2 (negative controls) were specifically tested for their ability to inhibit the activity of β-lactamases from *Escherichia coli* (TEM-1) and *Enterobacter cloacae*. Our results indicated that the RRM concept was successfully applied in the design of β-lactamase inhibitor peptide based on the frequency and phase of a particular enzyme. This study serves as a proof-of-concept to design peptide inhibitors against any specific β-lactamase class and provides the fundamental preliminary data required to determine the efficacy of pep3 peptide as β-lactamase inhibitor using clinical multidrug-resistant bacteria.

## Materials and methods

### Resonant recognition model

The RRM is a biophysical, theoretical model that can analyze interactions between proteins and their targets, which could be other proteins, DNA, RNA, or small molecules. The RRM has been previously published in detail in a number of publications [28–31]. The RRM model is based on the findings that certain periodicities (frequencies) within the distribution of energy of delocalized electrons along the protein backbone are critical for macromolecule biological function and/or interaction with their targets. The distribution of delocalized electron energies is calculated by assigning each amino acid a specific physical parameter representing the energy of delocalized electrons of each residue. Consequently, the spectral characteristics of such energy distribution (signal) are calculated using the Fourier Transform. This means that the linear numerical signal representing the distribution of energies along the macromolecule is transformed into the frequency domain and is characterized by a number of different frequencies containing all information from the original signal. Comparing such spectra using the cross-spectral function for macromolecules, which are sharing the same biological function/interaction, it has been shown that they share the same frequency within the spectrum of free energy distribution along the macromolecule [28–31]. Peak frequencies in such multiple cross-spectral functions present common frequency components for all macromolecular sequences compared. The comprehensive analysis done so far confirms that all macromolecular sequences, with a common biological function and/or interaction, have a common

frequency component, which is a specific feature for the observed biological function/interaction [28–31]. Thus, each specific macromolecular biological function/interaction within the macromolecule is characterized by a specific RRM frequency.

Each biological function is driven by proteins that selectively interact with other proteins, DNA/RNA regulatory segments, or small molecules. Through extensive use of the RRM model, it has been shown that proteins and their targets share the same matching RRM characteristic frequency [28–31]. The matching of periodicities within the distribution of energies of free electrons along the interacting proteins can be regarded as resonant recognition and as such is highly selective. Thus, the RRM frequencies characterize not only protein function, but also recognition and interaction between protein and its targets: proteins (receptors, binding proteins, and inhibitors), DNA/RNA regulatory segments, or small molecules. In addition, it has been also shown that interacting macromolecules have opposite phases at their characteristic RRM recognition frequency [28–31]. Every frequency can be presented by one sinusoid characterized by its three parameters: frequency, amplitude, and phase. The phase is presented in radians (rad) and can be between –π and +π (-3.14 and +3.14). The phase difference of or about π (3.14) is considered to be the opposite phase. The phase value can be presented in the phase circle where it is visually easier to observe phase differences (Fig 1).

## Bioactive peptide design

Once the characteristic frequency for the biological function of the protein is identified, it is possible to design new peptides/proteins with desired frequency components and

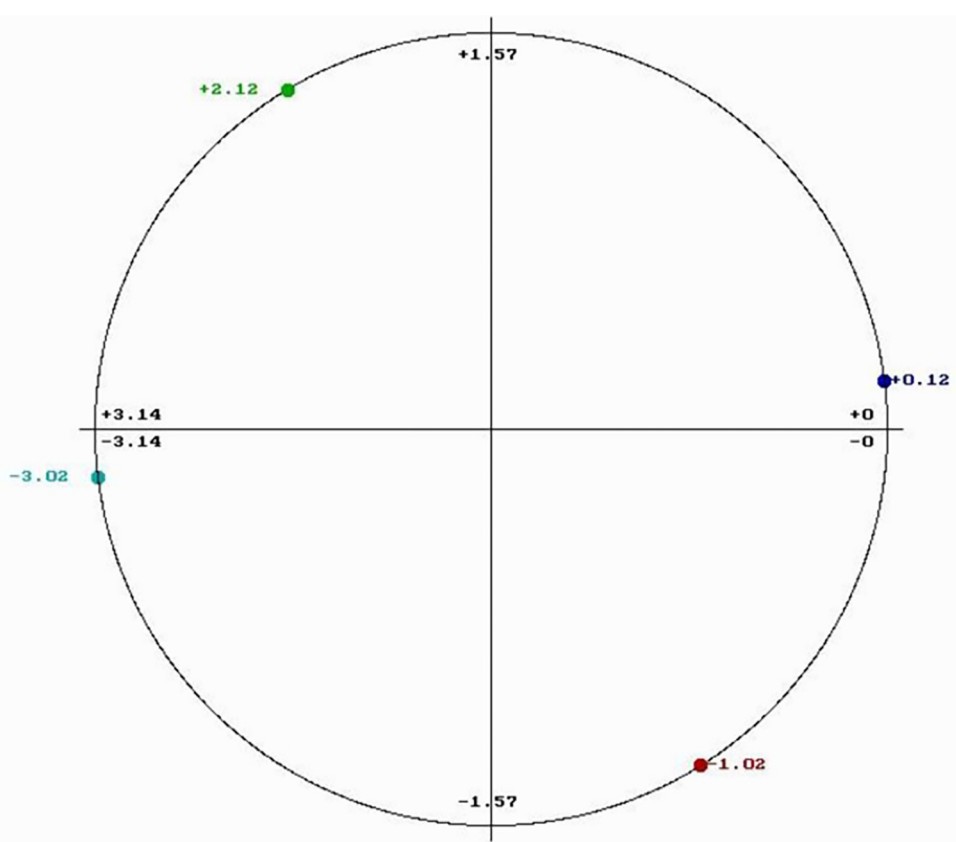

**Fig 1. Phase circle with phases at frequency f1 = 0.0352 chosen for the design of peptides.**

consequently with desired biological functions [28–30,32–38]. The process of bioactive peptide design is as follows: (1) Determination of RRM characteristic frequency using multiple cross-spectral functions for a group of protein sequences that share common biological function/interactions. (2) Determination of phases for the characteristic frequencies of a particular protein which is selected as the parent for agonist/antagonist peptide. (3) Calculation using Inverse Fourier Transform of the signal with characteristic frequency and phase. The minimal length of the designed peptide is defined by the characteristic frequency f as 1/f. (4) Determination of resulting amino acid sequence using tabulated EIIP parameter values.

This design approach has already been successfully applied and experimentally tested in the design of FGF analog [33], HIV envelope protein analog [36,38] peptide to mimic myxoma virus oncolytic function [32,35] as well as recently developed peptide that can prevent SARS-CoV-2 virus entry into the host cells via ACE2 receptor [34,37].

## Peptide synthesis and preparation

The *de novo* designed peptides (pep1, pep2, pep3, and pep4) were commercially synthesized by GenScript USA Inc, (New Jersey, USA) with ≥95% purity (Table 1). Aliquots of the lyophilized peptides were stored at -20˚C. Whenever needed, pep3 was dissolved in water (≤5 mg/ml) whereas peptides pep1, pep2, and pep4 were dissolved in dimethyl sulfoxide (DMSO, ≤5 mg/ml, ≤5 mg/ml, and ≤10 mg/ml respectively). Freshly prepared stocks in water or DMSO were used at the required concentration within the same day of preparation.

## β-lactamase activity assay

The β-lactamase activity was assayed by using a β-lactamase activity assay kit (Sigma-Aldrich, USA, catalog number MAK221). This assay is based on the hydrolysis of a non-antimicrobial, chromogenic cephalosporin called nitrocefin. Hydrolysis of nitrocefin by β-lactamase enzyme produces a colorimetric product with an absorbance maximum at 490 nm proportional to the enzymatic activity present. The amount of enzyme required to hydrolyze 1.0 μmol of nitrocefin per minute at pH 7.0 at 25˚C is equal to one unit of β-lactamase. Assay was done in a 96-well plate using commercially available TEM-1 β-lactamase from *E. coli* (catalog number PV3575, Thermo Fisher Scientific) and β-lactamase from *E. cloacae* (catalog number P4524-100 UN, Sigma-Aldrich) in the presence or absence of different concentrations of peptides pep1, pep2 and pep3 (as required). 50 μl of the unknown samples (assay buffer, enzyme, and peptide) were added to wells of a clear flat bottom 96-well plate and supplemented with a reaction mixture containing nitrocefin and assay buffer to a final volume of 100 μl. As needed, 2 μl of 1:100 diluted TEM-1 (10 ng/13.76 nmoles) and 0.0002 units of *E. cloacae* β-lactamase were used per well for the assay. Immediately after the addition of nitrocefin, absorbance at 490 nm was measured every minute, for 10 minutes at room temperature using a microplate reader (xMark Microplate Reader, BioRad).

**Table 1. List of *de novo* designed peptides pep1, pep2, pep3 and pep4.**

| Peptide name | Frequency | Phase (radian) | Solubility |
|:---:|:---:|:---:|:---:|
| Pep1 | 0.0352 | -3.02 | DMSO |
| Pep2 | 0.0352 | +2.12 | DMSO |
| Pep3 | 0.0352 | +0.12 | Water |
| Pep4 | 0.0352 | -1.02 | DMSO |

### Statistical analysis

All assays were performed in triplicate for each condition, and repeated at least three times unless otherwise stated. Error bars represent the standard deviations from the means. Statistical analysis was performed using two-tailed paired Student's t-test.

## Results

### RRM analysis of β-lactamase proteins

The RRM model was used to analyze 22 β-lactamase protein sequences (P62593, P9WKD3, A5U493, P52663, Q9S169, Q47066, P9WKD2, P0A5I7, O07293, P23954, Q9S424, P28585, P22391, Q06778, Q51574, O08337, P0A3M2, P96348, Q48406, Q9R976, Q93LM8, and P37321) representing several bacterial strains, from the UniProt database. As shown in Fig 2, a common RRM characteristic frequency (f1 = 0.0352±0.0041) was identified.

### Peptide design

The common RRM characteristic frequency may likely correlate with a similar protein function. Thus, a peptide designed with an opposite phase should be able to block β-lactamase activity and consequently interfere with the process of antibiotic inactivation. As shown in Table 2, the β-lactamases from the different bacterial types and strains had phase variations at RRM frequency f1 = 0.0352. However, most phases at this frequency are either clustered around the phase of -3.02 rad (as highlighted in blue) or the phase of +2.12 rad (highlighted in yellow). For a peptide design that could potentially block β-lactamase activity, would require phases that are opposite to the most prevalent phases for the different bacterial types and strains: +0.12 rad as opposite to -3.02 rad and -1.02 rad as opposite to +2.12 rad. Because the RRM frequency is f1 = 0.0352, the minimum length of designed peptides is predicted to be 1/f1 (1/0.0352 = 28.4). Thus, we choose to design four 30-mer peptides with frequency f1 = 0.0352 and phases: -3.02 rad (pep1), +2.12 rad (pep2), +0.12 rad (pep3) and -1.02 rad (pep4) respectively (Table 1). The chosen phases are presented in a phase circle in Fig 1. Peptide pep3 is expected to block the β-lactamase activity of all enzymes highlighted in blue in Table 2 (predominantly from *E. coli* and *Klebsiella*) whereas peptide pep4 is supposed to block the activity of yellow highlighted enzymes in Table 2 (predominantly from *Mycobacterium tuberculosis* and *Pseudomonas aeruginosa*). Peptides pep1 and pep2 are negative controls for pep3 and pep4 respectively.

β-lactamases from *E. coli* (TEM-1), *E. cloacae* (representatives of blue highlighted enzymes) and β-lactamase from *P. aeruginosa* (to represent yellow highlighted enzymes) were selected to

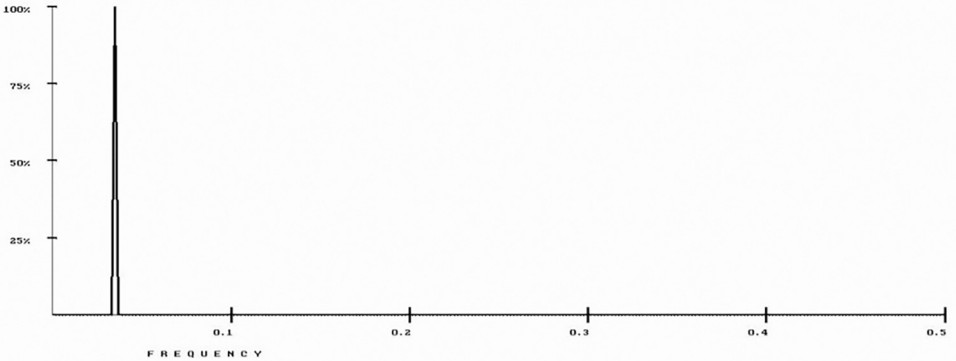

**Fig 2. RRM cross-spectrum of β-lactamase protein sequences with common RRM characteristic frequency at f1 = 0.0352±0.0041.**

**Table 2. Phases for β-lactamases from different bacterial strains.**

| Bacterial strains with β-lactamases | Phase at f1 = 0.0352 |
|---|---|
| *Mycobacterium tuberculosis* BLAC-MYCTA (BlaC) | +2.12 |
| *Pseudomonas aeruginosa* BLO18-PSEAI | -0.01 |
| *Escherichia coli* BLAB-ECOLX (BlaB) | -3.08 |
| *Pseudomonas aerudino* BLA5-PSEAI (SHV5) | -3.02 |
| *Mycobacterium bovis* BLAC-MYCBO (BlaC) | -2.83 |
| *Klebsiella oxytoca* BLO1-KLEOX (Oxa-1) | +2.22 |
| *Klebsiella oxytoca* BLO2-KLEOX (Oxa-2) | +3.07 |
| *Escherichia coli* BLC1-ECOLX (CTX-M-1) | -3.12 |
| *Pseudomonas aeruginosa* BLE1-PSEAI (Per-1) | +2.49 |
| *Enterobacter cloacae* BLAN-ENTCL (NmcA) | +3.01 |
| *Escherichia coli* BLAT-ECOLX (TEM-1) | -3.01 |
| *Klebsiella pneumoniae* BLA6-KLEPN (SHV-6) | -3.07 |
| *Mycobacterium tuberculosis* BLAC-MYCTO (BlaC) | +2.12 |
| *Mycobacterium tuberculosis* BLAC-MYCTU (BlaC) | +2.12 |
| *Escherichia coli* BLT1-ECOLX (Toho-1) | -2.66 |
| *Klebsiella oxytoca* BLAT-KLEOX (TEM-12) | -3.00 |
| *Pseudomonas aeruginosa* BLO15-PSEAI | -1.58 |
| *Escherichia coli* BLA34-ECOLX (SHV-34) | -3.02 |
| *Pseudomonas aeruginosa* BLO19-PSEAI (Oxa-19) | +2.17 |
| *Escherichia coli* BLA24-ECOLX (SHV-24) | -3.08 |
| *Klebsiella pneumoniae* BLA13-KLEPN (SHV-13) | -2.99 |
| *Pseudomonas aeruginosa* BLO11-PSEAI (Oxa-11) | +1.93 |

It can be observed that most phases are either clustered around phase of -3.02 rad (highlighted in blue) or phase of +2.12 rad (highlighted in yellow).

test the inhibitory efficiencies of pep3 and pep4 as these are the only enzymes available commercially from Table 2. Due to some solubility issues of pep4 in DMSO, it could not be used in experiments with *P. aeruginosa* β-lactamase and therefore only results of pep3 testing with *E. coli* and *E. cloacae* enzymes are shown below.

## Effect of pep3 peptide on *E. coli* TEM-1 β-lactamase activity

To evaluate the ability of the RRM-derived peptides to inhibit β-lactamase activity, the efficacy of the TEM-1 β-lactamase from *E. coli* was determined in the presence of different concentrations of pep3 peptide. As shown in Fig 3, 200 μg of the pep3 had no effect on the β-lactamase activity of TEM-1. At higher concentrations (300 and 400 μg), pep3 decreased the activity by 50% and ~90% respectively whereas 500 μg of pep3 completely inhibited the activity of TEM-1 (Fig 3A and 3B). Since 500 μg of pep3 fully abolished the TEM-1 activity, 500 μg of peptides pep1 and pep2 were used as negative controls using the same assay. Similar to TEM-1 in the absence of any peptide, pep1 and pep2 at a concentration of 500 μg did not inhibit the TEM-1 β-lactamase activity (Fig 4A and 4B). Taken together, these data suggest that pep3 can be a specific inhibitor for the TEM-1 β-lactamase enzyme.

## Effect of pep3 peptide on β-lactamase activity from *E. cloacae*

The ability of pep3 peptide to function as β-lactamase inhibitor, was also evaluated by using the β-lactamase enzyme from *E. cloacae* in the presence of different concentrations of pep3. As

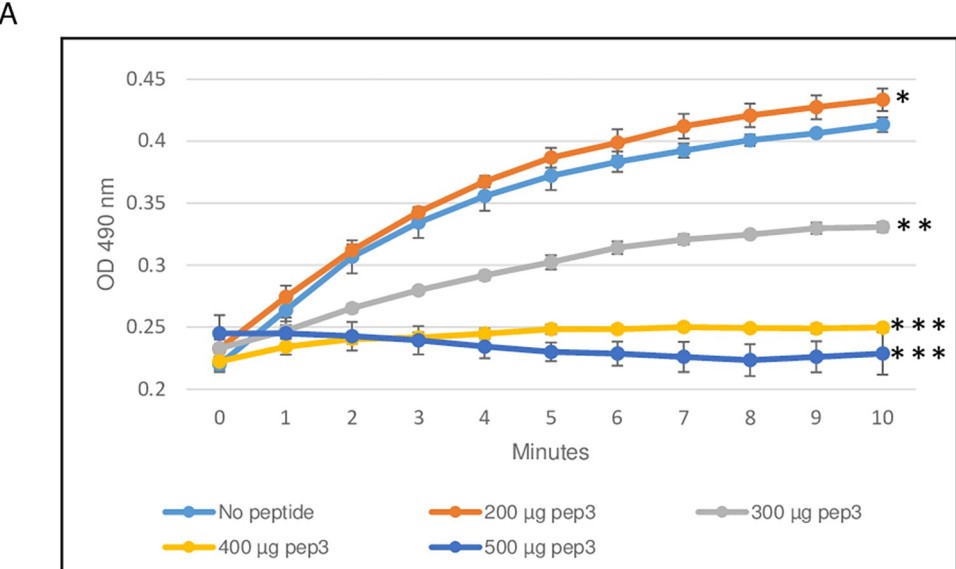

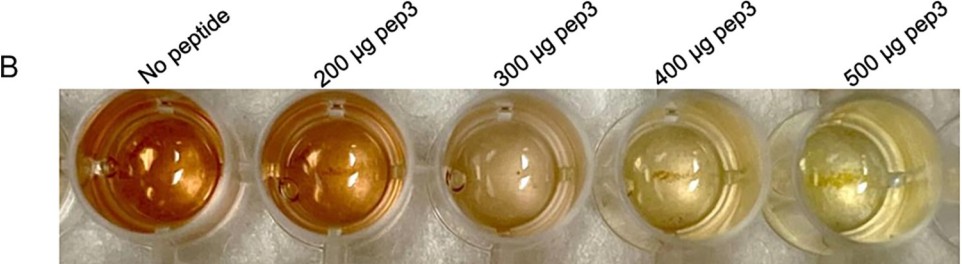

**Fig 3. β-lactamase activity assay of *E. coli* TEM-1 using 200–500 μg of pep3 peptide.** (A) The assay was performed in 96-well plate in a 100 μl total reaction volume containing assay buffer, TEM-1 β-lactamase and nitrocefin in absence or presence of pep3 peptide. The reactions were followed by measuring absorbance at 490 nm for 10 minutes with 1-minute interval. The results represent the means of three independent experiments. Error bars represent the standard deviations from the means. (B) Brown color in well with no peptide is due to hydrolyzed nitrocefin which rapidly changes color from yellow to brown when degraded due to hydrolysis. Statistical analysis was performed using two-tailed paired Student's t-test (*, $p < 0.2$; **, $p < 0.01$; ***, $p \leq 0.001$ vs. no peptide control).

shown in Fig 5, 100 μg of the pep3 peptide had no effect on the β-lactamase activity of *E. cloacae*. Using higher concentrations of 200 and 300 μg of pep3, the activity was decreased by ~50% and ~90% respectively. 400 μg of pep3 completely inhibited the *E. cloacae* enzyme activity (Fig 5A and 5B). As negative controls, 400 μg of pep1 and pep2 did not significantly affected the *E. cloacae* β-lactamase activity (Fig 6A and 6B). Taken together, the data suggest that pep3 may function as a specific inhibitor for the *E. cloacae* β-lactamase enzyme.

## Discussion

The RRM model has already been used as an established tool for the *de novo* design of short bioactive peptides that can modulate various biological functions [32–39]. For example, an 18-mer IL12 peptide was able to induce cytotoxic effects on the B16F0 mouse melanoma cell line [39] and the RRM-MV peptide (2.3 kDa, 18-mer) was successfully tested as a potential cancer therapeutic agent *in vitro* using tumor and normal cell lines [35]. Recently, this

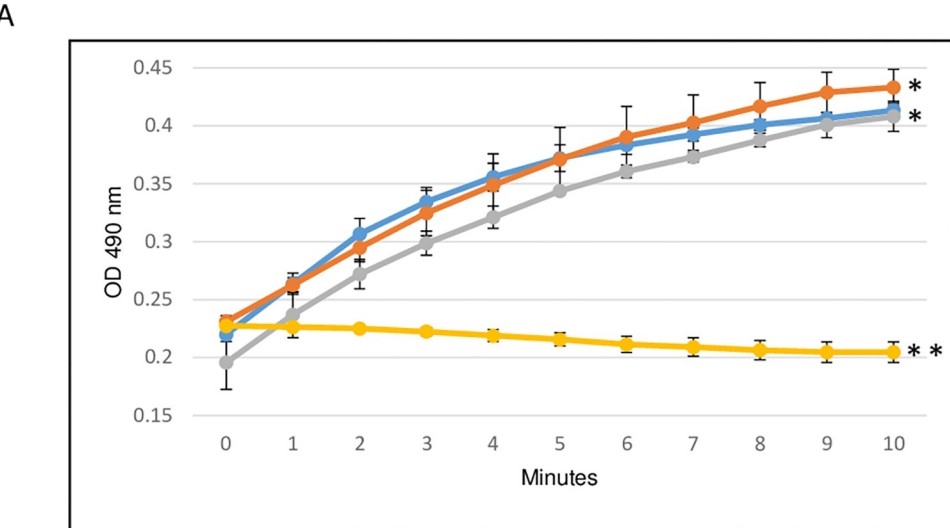

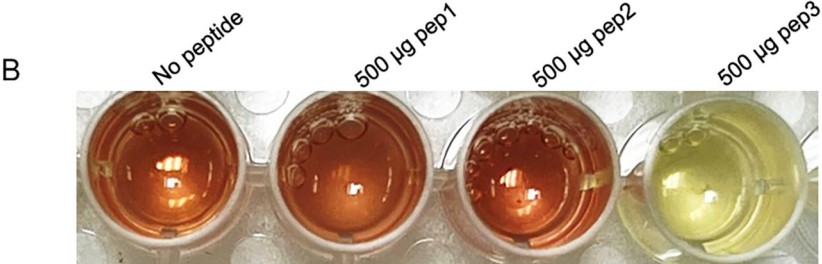

**Fig 4. β-lactamase activity assay of *E. coli* TEM-1 using 500 μg of pep1, pep2 and pep3 peptides.** (A) The assay was performed in 96-well plate in a 100 μl total reaction volume containing assay buffer, TEM-1 β-lactamase and nitrocefin in absence or presence of peptides pep1, pep2 and pep3. The reactions were followed by measuring absorbance at 490 nm for 10 minutes with 1-minute interval. The results represent the means of three independent experiments. Error bars represent the standard deviations from the means. (B) Brown color in wells with no peptide, 500 μg of pep1 and 500 μg of pep2 is due to hydrolyzed nitrocefin which rapidly changes color from yellow to brown when degraded due to hydrolysis. Statistical analysis was performed using two-tailed paired Student's t-test (*, $p < 0.5$; **, $p < 0.005$ vs. no peptide control).

approach has been used to design a CovA peptide (as a covid-19 drug candidate) that can prevent the entry of the SARS-CoV-2 virus into the host cells via Angiotensin-Converting Enzyme 2 (ACE2) receptor [37].

In the current study, we have employed the RRM model to design peptides with the likely capability of blocking β-lactamase activity and preventing antimicrobial resistance (AMR). Bacterial resistance to β-lactam antibiotics is one of the leading global public health threats of the current century [12,40]. Although significant efforts have been made in the last few years to deal with different aspects of antibiotic resistance, the continuous evolution of new variants resistant to newer β-lactams and/or β-lactamase inhibitors still poses a challenge to global AMR threat. Using the RRM model, we have analyzed several β-lactamase protein sequences (representing the most widely distributed class A and D enzymes) to calculate their characteristic RRM frequency. Interestingly, there was only one common characteristic frequency that clustered around two phases for all analyzed proteins from different bacteria representing different β-lactamase activity classes. It is likely that the common characteristic frequency may be

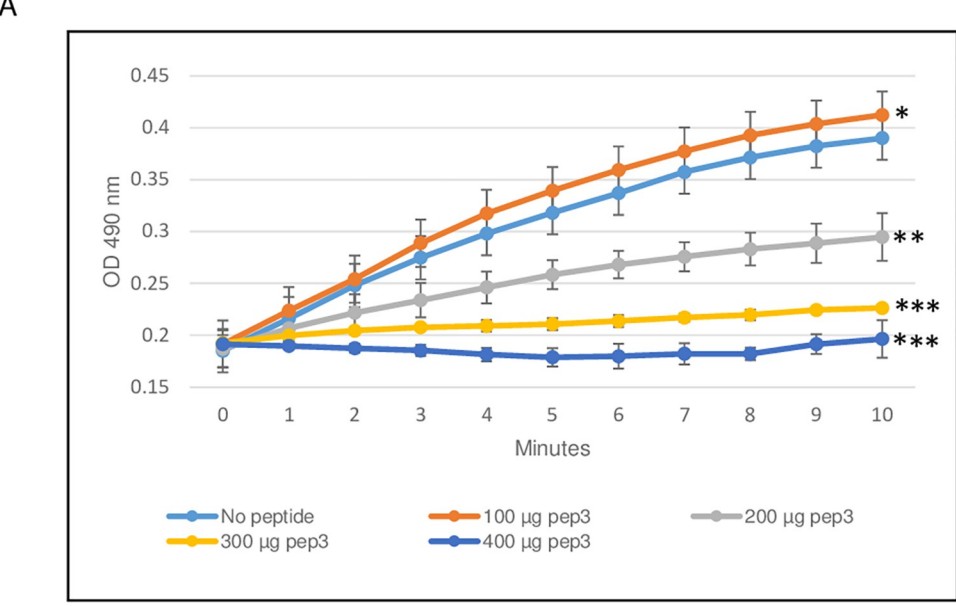

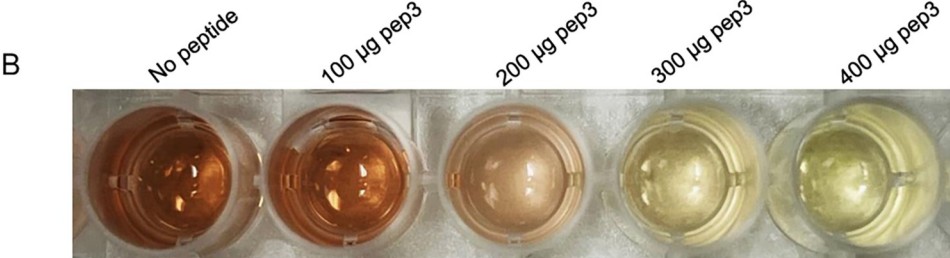

**Fig 5. β-lactamase activity assay of *E. cloacae* using 100–400 μg of pep3 peptide.** (A) The assay was performed in 96-well plate in a 100 μl total reaction volume containing assay buffer, *E. cloacae* β-lactamase and nitrocefin in absence or presence of pep3 peptide. The reactions were followed by measuring absorbance at 490 nm for 10 minutes with 1-minute interval. The results represent the means of three independent experiments. Error bars represent the standard deviations from the means. (B) Brown color in well with no peptide is due to hydrolyzed nitrocefin which rapidly changes color from yellow to brown when degraded due to hydrolysis. Statistical analysis was performed using two-tailed paired Student's t-test (*, $p = 0.5$; **, $p < 0.05$; ***, $p \leq 0.005$ vs. no peptide control).

related to the similar mechanistic function of the proteins. Inhibitor peptides designed with opposite phases were predicted to modulate the activity of their respective β-lactamase enzymes. Inhibition of *E. coli* and *E. cloacae* β-lactamases by pep3 peptide ($p \leq 0.001$ and $p \leq 0.005$ vs. no peptide control respectively) confirmed the functionality of this model to successfully design a β-lactamase inhibitor. The negative control peptide (pep1), with a similar phase to the TEM-1/*E. cloacae* β-lactamase, did not affect the activity which may suggest that pep3 in the opposite phase is target specific and can modulate protein function. Because of the commercial unavailability of other β-lactamase enzymes with a similar pep3 phase (highlighted in blue, Table 2), its effect on their activity is unknown. Experiments are ongoing in the laboratory to clone and purify other members from Table 2 to evaluate their activities in the presence of the pep3 inhibitor peptide. Different studies to see the effect of this peptide on multiple AMR-resistant clinical strains of *E. coli* and *E. cloacae* are also ongoing in the laboratory.

Carbapenems are broad spectrum antibiotics that are effective against most β-lactamases including MBLs and extended spectrum β-lactamases. Because they are the last-resort

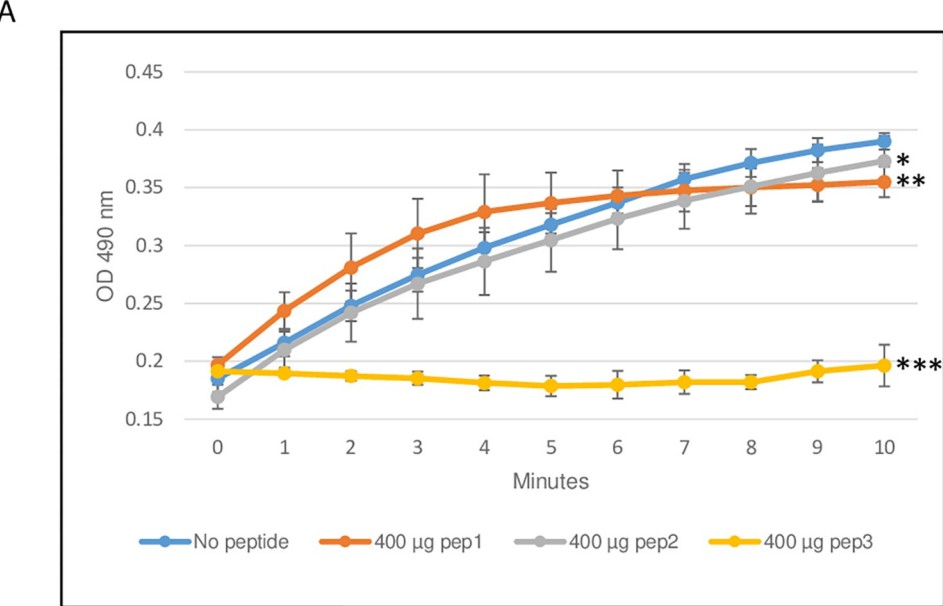

A

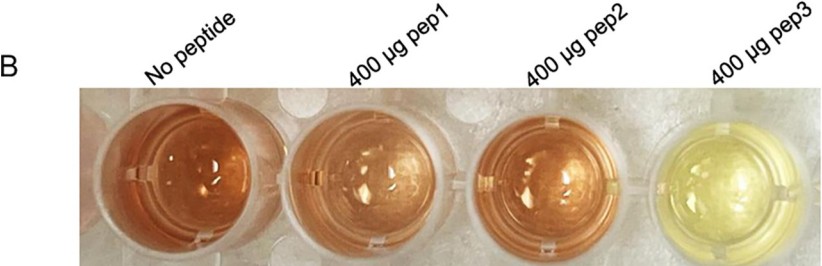

B

**Fig 6. β-lactamase activity assay of *E. cloacae* using 400 μg of pep1, pep2 and pep3 peptides.** (A) The assay was performed in 96-well plate in a 100 μl total reaction volume containing assay buffer, *E. cloacae* β-lactamase and nitrocefin in absence or presence of peptides pep1, pep2 and pep3. The reactions were followed by measuring absorbance at 490 nm for 10 minutes with 1-minute interval. The results represent the means of three independent experiments. Error bars represent the standard deviations from the means. (B) Brown color in wells with no peptide, 400 μg of pep1 and 400 μg of pep2 is due to hydrolyzed nitrocefin which rapidly changes color from yellow to brown when degraded due to hydrolysis. Statistical analysis was performed using two-tailed paired Student's t-test (*, $p < 0.5$; **, $p < 0.05$; ***, $p \leq 0.005$ vs. no peptide control).

antibiotics for many infections, resistance to carbapenems, specifically against gram-negative bacteria, is an urgent health concern [41]. Based on the critical need to develop new antibiotics, the WHO in 2017 published a list of antibiotic-resistant "priority pathogens" (critical, high, and medium) [42]. The "critical priority" pathogens included carbapenem-resistant *Acinetobacter baumannii* and *P. aeruginosa* as well as carbapenem-resistant extended-spectrum β-lactamase-producing Enterobacteriaceae including *Klebsiella*, *E. coli*, *Serratia*, and *Proteus* [42]. They are resistant to carbapenem due to the production of various carbapenemases including KPC, IMP, VIM, NDM and OXA enzymes [14,43]. These enzymes can hydrolyze carbapenems, cephalosporins and penicillins which limits the treatment options against gram-negative bacteria [5,44]. Newer variants of carbapenemases are constantly emerging, such as, the KPC-55 variant, which inactivates aztreonam and meropenem [45], and NDM-19 which can function in the presence of low concentrations of zinc [46]. Among different classes of β-lactamases, Class B MBLs are most challenging due to their ability to hydrolyze and develop

resistance to nearly all existing β-lactam antibiotics and the unavailability of clinically useful drug regimens for MBLs [27,47,48]. To date, there are no clinically available inhibitors for MBLs. Research on carbapenem-resistant "critical priority" pathogens and MBLs needs to be done urgently in a prioritized way with an aim to develop new antimicrobial strategies [42,49] including combinations of β-lactams with new β-lactamase inhibitors which exhibit antibacterial effects against carbapenemases and MBL producing bacteria. We propose to use the RRM model to design peptide inhibitors against above-mentioned carbapenemases and MBLs. Research is in progress in our laboratory to analyze the protein sequences of (1) representative carbapenemase from *A. baumannii*, *P. aeruginosa*, *K. pneumoniae*, *E. coli* and (2) class B MBLs (including NDM, VIM, IMP, SPM types) to identify their characteristic RRM frequency and phases (data not published). The RRM approach can be used to design β-lactamase inhibitor peptides for both broad spectrum enzymes as well as against a particular class due to the advantage that it does not look into structural characteristics of the binding domain but uses the full protein's biophysical parameters that are important for its binding activity.

In conclusion, antibiotic resistance is an important universal health threat showing resistance to most of the clinically used antibiotics [40]. Here, we have used the RRM model to design a novel 30-mer peptide that inhibited the activity of class A β-lactamases from *E. coli* (TEM-1) and *E. cloacae*. The results presented here could provide the basis for the development of new antimicrobial drugs and support the use of the RRM approach to increase the efficacy of β-lactam antibiotics for treating infections caused by multidrug-resistant bacteria.

## Author Contributions

**Conceptualization:** Arunima Mishra, Irena Cosic, Ivan Loncarevic, Drasko Cosic, Hansel M. Fletcher.

**Formal analysis:** Arunima Mishra.

**Funding acquisition:** Arunima Mishra, Ivan Loncarevic, Hansel M. Fletcher.

**Investigation:** Arunima Mishra.

**Methodology:** Irena Cosic, Drasko Cosic.

**Project administration:** Arunima Mishra.

**Resources:** Irena Cosic, Ivan Loncarevic, Drasko Cosic, Hansel M. Fletcher.

**Supervision:** Hansel M. Fletcher.

**Validation:** Arunima Mishra.

**Visualization:** Arunima Mishra.

**Writing – original draft:** Arunima Mishra, Irena Cosic.

**Writing – review & editing:** Arunima Mishra, Irena Cosic, Ivan Loncarevic, Drasko Cosic, Hansel M. Fletcher.

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
