## [Decision Letter · Decision Letter 0]

17 Jul 2023

PONE-D-23-14582Inhibition of β-lactamase function by de novo designed peptidePLOS ONE

Dear Dr. Mishra,

Thank you for submitting your manuscript to PLOS ONE. After careful consideration, we feel that it has merit but does not fully meet PLOS ONE’s publication criteria as it currently stands. Therefore, we invite you to submit a revised version of the manuscript that addresses the points raised during the review process.

We look forward to receiving your revised manuscript.

Kind regards,

Farah Al-Marzooq, MD, PhD

Academic Editor

PLOS ONE

https://pubmed.ncbi.nlm.nih.gov/36193979/

https://journals.plos.org/plosone/article?id=10.1371%2Fjournal.pone.0024809

https://www.nature.com/articles/s41579-022-00820-y?code=8d784f4b-3b18-4ec2-b365-0caab873a026&error=cookies_not_supported

In your revision ensure you cite all your sources (including your own works), and quote or rephrase any duplicated text outside the methods section. Further consideration is dependent on these concerns being addressed.

Additional Editor Comments:

Please try to revise the manuscript critically and consider all the comments of the reviewers

Reviewers' comments:

Reviewer's Responses to Questions

**Comments to the Author**

1. Is the manuscript technically sound, and do the data support the conclusions?

Reviewer #1: Yes

Reviewer #2: Partly

2. Has the statistical analysis been performed appropriately and rigorously? 

Reviewer #1: Yes

Reviewer #2: No

3. Have the authors made all data underlying the findings in their manuscript fully available?

Reviewer #1: Yes

Reviewer #2: No

4. Is the manuscript presented in an intelligible fashion and written in standard English?

Reviewer #1: Yes

Reviewer #2: Yes

5. Review Comments to the Author

Reviewer #1: Reviewer Comments:

-Abstract: Please add background/problem statement and objectives. The abstract did not reflect the main findings of the research. Why resistance to β-lactam antibiotics was not address correctly.

Introduction: is good in story line however more information need to be addressed what cause the resistance to the antibiotic. In line 64 the authors mention (multidrug-resistance) what does mean? must explain it ? and how bacteria developed multi drug resistance? In line 71 the mechanisms of resistance to β-lactam antibiotics was not address adequately.

In line 98 Why chose Resonant Recognition Model (RRM) this is not address in Introduction section as well.The mechanism of action of this inhibitor was not address adequately.

Methodology: comprehensive and adequate but the paragraph 143-178must be mention in introduction not suitable in methods

In line 186 Determination of RRM characteristic frequency using multiple cross-spectral functions for a group of protein sequences that share common biological function/interactions. Why authors use multiple cross spectral functions need more explain?

In line 215 the author must mention the source or of bacteria (specimen) that used for isolation of β-lactamase? Did the authors used pathogenic bacteria for obtained β-lactamase?

In line 203 please define the abbreviation DMSO

In line 233 Statistical analysis. The authors must be mention the type of analysis and version of program that used in statistical analysis

Result

Why pep3 peptide had inhibition effect on on the β lactamase activity of TEM-1 while pep1 and pep2 had no effect why pep3 have more inhibition effect

The authors must be compare between the effect of pep3 on β-lactamase the isolated from E.coli and E. cloacae

-Quality of images was low?

Discussion

Interestingly, Using the RRM model, we have analyzed several β-lactamase protein sequences (representing the most widely distributed class A and D enzymes) to calculate their characteristic RRM frequency. Why the authors chose class A and D enzymes not class B metallo-β-lactamases (MBLs and as we know none of the clinically approved inhibitors can effectively inhibit the members of this class

In line 358-360 Inhibition of E. coli 359 and E. cloacae β-lactamases by pep3 peptide confirmed the functionality of this model to 360 successfully design a β-lactamase inhibitor. Statistical analysis is needed.Is there significant association ?p value not mention?

Conclusion

Author must address whether resistance to B- lactam antibiotics an alarming situation? conclusion is not addressed well

Many references are OLD one, please include the latest especially on Introduction and Discussions

1- Hussein, RA, Al-Ouqaili MTS, Majeed YH. (2022). Detection of clarithromycin resistance and 23SrRNA point mutations in clinical isolates of Helicobacter pylori isolates: Phenotypic and molecular methods, Saudi Journal of Biological Sciences, 29 (1).

2- AL-KUBAISY SH, HUSSEIN, RA, AL-OUQAILI, MTS. (2020). Molecular Screening of Ambler class C and extended spectrum β-lactamases in multi-drug resistant Pseudomonas aeruginosa and selected species of Enterobacteriaceae. International Journal of Pharmaceutical Research | Jul - Sep 2020 | Vol 12 | Issue 3

3- Al-Ouqaili, MTS, Al-Taei, SA, Al-Najjar A. Molecular Detection of Medically Important Carbapenemases Genes Expressed by Metallo-β-lactamase Producer Isolates of Pseudomonas aeruginosa and Klebsiella pneumoniae. Asian Journal of Pharmaceutics • Jul -Sep 2018 (Suppl ) • 12 (3) | S991

4- Khalaf, EA, Al-Ouqaili, MTS. Molecular detection and sequencing of SHV gene encoding for extended-spectrum β-lactamases produced by multidrug resistance some of the Gram-negative bacteria. International Journal of Green Pharmacy • Oct-Dec 2018 (Suppl) • 12 (4) | S910-S918.

5- Al-Qaysi, A. K., Al-Ouqaili, . M. T. & Al-Meani, . S. A. (2020). Ciprofloxacin- and gentamicin-mediated inhibition of Pseudomonas aeruginosa biofilms is enhanced when combined the volatile oil from Eucalyptus camaldulensis. SRP, 11 (7), 98-105.

Reviewer #2: This manuscript describes the urgency of developing new antibacterial treatments to combat antimicrobial resistance, especially for multidrug-resistant bacteria. The authors' investigation into the Resonant Recognition Model (RRM) to design 30-mer peptides as β-lactamase inhibitors is detailed, showing promising results with 100% inhibition of class A β-lactamases. This approach presents a potential solution to address antimicrobial resistance by allowing the design of inhibitors for specific classes of bacteria. However, the manuscript lacks sufficient details and explanations, raising concerns about the following points:

1.The choice of 30-mer peptides as inhibitors is not clarified. It is important to explain why this specific length was selected over shorter or longer peptide sequences and what advantages it offers.

2.The possible toxicity of the designed peptides is not addressed. It is essential to investigate and discuss any potential toxic effects that these inhibitors may have on human cells or beneficial bacteria.

3.The manuscript does not explain how the peptides can avoid hydrolysis. Elaborating on methods that can be employed to prevent degradation will enhance the credibility of the proposed approach.

4.The sequences of the 30-mer peptides are not provided. It is crucial to include this information to allow other researchers to replicate and validate the findings.

5.The writing lacks sufficient explanations, which can make it difficult for readers to understand the methodology and results. Adding more detailed explanations, experimental procedures, and data interpretations will improve the clarity and impact of the manuscript.

Addressing these concerns will strengthen the manuscript and make it more informative and impactful for the scientific community.

6. PLOS authors have the option to publish the peer review history of their article (what does this mean?). If published, this will include your full peer review and any attached files.

Reviewer #1: **Yes: **Rawaa A. Hussein

Reviewer #2: No

---

## [Author Response · Author response to Decision Letter 0]

3 Aug 2023

PONE-D-23-14582

Inhibition of β-lactamase function by de novo designed peptide

PLOS ONE

The authors wish to thank the reviewers for their constructive comments. In the revised version of the manuscript, we have addressed all the questions and comments raised in the critique. The reviewers’ comments are given in black and the reply is given below each query in red. The corrections made in the manuscript are highlighted in yellow. The page and line numbers given in the response refer to the revised version.

Journal requirements

We have ensured that our manuscript meets PLOS ONE's style requirements, including those for file naming.

https://pubmed.ncbi.nlm.nih.gov/36193979/

https://journals.plos.org/plosone/article?id=10.1371%2Fjournal.pone.0024809

https://www.nature.com/articles/s41579-022-00820-y?code=8d784f4b-3b18-4ec2-b365-0caab873a026&error=cookies_not_supported

In your revision ensure you cite all your sources (including your own works), and quote or rephrase any duplicated text outside the methods section. Further consideration is dependent on these concerns being addressed.

We have critically rephrased any duplicated texts (page 4, lines 71-73; page 18, lines 345-349; page 19, lines 378-382 and 385-391) and added the relevant citation (reference 15).

We did not include the detailed sequences of peptides pep1-4 in the manuscript due to unresolved intellectual property challenges. We are willing to share the sequence in response to individual requests (to Hansel Fletcher: hfletcher@llu.edu and/or Ivan Loncarevic: ivlon@ymail.com).

Reviewer #1: Reviewer Comments

-Abstract: Please add background/problem statement and objectives. The abstract did not reflect the main findings of the research. Why resistance to β-lactam antibiotics was not address correctly.

We have modified the abstract as per reviewer suggestions (page 2, lines 26, 28-31, 37-38).

Introduction: is good in story line however more information need to be addressed what cause the resistance to the antibiotic. In line 64 the authors mention (multidrug-resistance) what does mean? must explain it ? and how bacteria developed multi drug resistance? In line 71 the mechanisms of resistance to β-lactam antibiotics was not address adequately.

Multidrug resistance means resistance to two or more drugs. This information was already included in the submitted manuscript (page 4, lines 68-69). For clarity we have underlined these lines.

The mechanisms of resistance to β-lactam antibiotics is now modified in the revised version as per reviewer suggestions (page 5, lines 78-83).

In line 98 Why chose Resonant Recognition Model (RRM) this is not address in Introduction section as well. The mechanism of action of this inhibitor was not address adequately.

Earlier we have included this information in the discussion. We have now modified the Introduction to explain the advantages of RRM model (page 6, lines 117-120). The RRM model has been chosen here to analyse the activities of beta-lactamase, as it is based on completely new aspects of protein properties and thus can introduce new better way of fighting against beta-lactamase bacterial resistance. The RRM approach can be used to design β-lactamase inhibitor peptides for both broad spectrum enzymes as well as against a particular class due to the advantage that it does not look into structural characteristics of the binding domain but uses the full protein’s biophysical parameters that are important for its binding activity.

The physical background of the RRM model has been explained in details in number of previous publications (28-31). In summary, interaction between proteins was found to be based on resonant energy transfer between interacting proteins at the specific frequency. Same principle of action/interaction is proposed for beta-lactamase inhibitor designed using the RRM model. This has been explained in detail in our methodology section.

Methodology: comprehensive and adequate but the paragraph 143-178must be mention in introduction not suitable in methods. 

This is the main method utilized for the study. Because this is the first application for this approach we deem it necessary to include this section in the method section of the manuscript. We have also included this in “Introduction” as necessary (page 6, lines 108-120).

In line 186 Determination of RRM characteristic frequency using multiple cross-spectral functions for a group of protein sequences that share common biological function/interactions. Why authors use multiple cross spectral functions need more explain?

Multiple cross-spectral function generally is used to compare number of different spectra to find out if there is any common frequency within analysed spectra. The peak(s) in such cross-spectral function denote common frequency(ies) for all analysed spectra. It has been found that spectra of protein sequences that have the same biological function/interaction have one unique peak in related cross-spectral function denoting one unique common frequency, which is found to characterise their common biological function/interaction. This finding is the basis of the RRM model (28-31).

In line 215 the author must mention the source or of bacteria (specimen) that used for isolation of β-lactamase? Did the authors used pathogenic bacteria for obtained β-lactamase?

We have used commercially available beta-lactamases for this study. The name of the company and catalog numbers were already mentioned in the submitted manuscript. To clarify, we have underlined these lines (page 11, lines 225-228). 

In line 203 please define the abbreviation DMSO

DMSO is defined in the revised manuscript (page 10, line 207).

In line 233 Statistical analysis. The authors must be mention the type of analysis and version of program that used in statistical analysis.

We have included the statistical analysis (type of analysis and program) in the revised manuscript and also added the p-values for figures 3, 4, 5 and 6 (page 12, lines 239-240; page 16, lines 301-303 and 312-313; page 17, lines 331-333 and 342-343).

Result

Why pep3 peptide had inhibition effect on the β lactamase activity of TEM-1 while pep1 and pep2 had no effect why pep3 have more inhibition effect

The authors must be compare between the effect of pep3 on β-lactamase the isolated from E. coli and E. cloacae

-Quality of images was low?

It has been explained in RRM methodology that proteins are supposed to interact if they have the same frequency and opposite phase at this frequency. Pep3 is designed to have same frequency and opposite phase to beta-lactamase and that is why it is proposed to interact and block the activity of beta-lactamase enzyme which is in complete agreement with the experimental results. On the other hand, for pep1 and pep2, although they have the same frequency, their phase at that frequency is not opposite and therefore they have no effect. Pep1 and pep2 are negative controls and therefore are not supposed to have any effect on E. coli and E. cloacae β lactamase enzymes. This has been explained in the manuscript as necessary (page 14, lines 266-267 and 269-270; page 15, lines 289-290; page 17, lines 321-322). For clarity, we have underlined these lines. This result once more confirms that not only frequency is important for the interaction but the phase at that characteristic frequency is also crucial.

We have compared the effects of pep3 on E. coli and E. cloacae beta-lactamase activity. Figure 3 describes the effect of pep3 on E. coli beta-lactamase activity whereas figure 5 explains the effect of pep3 on E. cloacae beta-lactamase activity.

We have improved the quality of images as per reviewer suggestion.

Discussion

Interestingly, Using the RRM model, we have analyzed several β-lactamase protein sequences (representing the most widely distributed class A and D enzymes) to calculate their characteristic RRM frequency. Why the authors chose class A and D enzymes not class B metallo-β-lactamases (MBLs and as we know none of the clinically approved inhibitors can effectively inhibit the members of this class).

RRM model has never been used to design inhibitors for any beta-lactamase class and therefore as a proof-of-concept to design peptide inhibitors, we have chosen some of the most prevalent and widely distributed class A and OXA enzymes. Now that our results have shown the functionality of this model, we have started research in our laboratory to analyze and design the beta-lactamase inhibitors for class B enzymes. The results will be published elsewhere.

In line 358-360 Inhibition of E. coli 359 and E. cloacae β-lactamases by pep3 peptide confirmed the functionality of this model to 360 successfully design a β-lactamase inhibitor. Statistical analysis is needed. Is there significant association ?p value not mention?

We have done the statistical analysis and added the p-values as per reviewer’s suggestions (page 19, line 367).

Conclusion

Author must address whether resistance to B- lactam antibiotics an alarming situation? conclusion is not addressed well

We have modified the conclusion as per reviewer’s suggestion (page 20, lines 408-409).

Many references are OLD one, please include the latest especially on Introduction and Discussions

1- Hussein, RA, Al-Ouqaili MTS, Majeed YH. (2022). Detection of clarithromycin resistance and 23SrRNA point mutations in clinical isolates of Helicobacter pylori isolates: Phenotypic and molecular methods, Saudi Journal of Biological Sciences, 29 (1).

2- AL-KUBAISY SH, HUSSEIN, RA, AL-OUQAILI, MTS. (2020). Molecular Screening of Ambler class C and extended spectrum β-lactamases in multi-drug resistant Pseudomonas aeruginosa and selected species of Enterobacteriaceae. International Journal of Pharmaceutical Research | Jul - Sep 2020 | Vol 12 | Issue 3

3- Al-Ouqaili, MTS, Al-Taei, SA, Al-Najjar A. Molecular Detection of Medically Important Carbapenemases Genes Expressed by Metallo-β-lactamase Producer Isolates of Pseudomonas aeruginosa and Klebsiella pneumoniae. Asian Journal of Pharmaceutics • Jul -Sep 2018 (Suppl ) • 12 (3) | S991

4- Khalaf, EA, Al-Ouqaili, MTS. Molecular detection and sequencing of SHV gene encoding for extended-spectrum β-lactamases produced by multidrug resistance some of the Gram-negative bacteria. International Journal of Green Pharmacy • Oct-Dec 2018 (Suppl) • 12 (4) | S910-S918.

5- Al-Qaysi, A. K., Al-Ouqaili, . M. T. & Al-Meani, . S. A. (2020). Ciprofloxacin- and gentamicin-mediated inhibition of Pseudomonas aeruginosa biofilms is enhanced when combined the volatile oil from Eucalyptus camaldulensis. SRP, 11 (7), 98-105.

We have updated the references with the latest and most relevant citations (reference numbers 12, 40 and 43). 

Reviewer #2: Reviewer Comments

This manuscript describes the urgency of developing new antibacterial treatments to combat antimicrobial resistance, especially for multidrug-resistant bacteria. The authors' investigation into the Resonant Recognition Model (RRM) to design 30-mer peptides as β-lactamase inhibitors is detailed, showing promising results with 100% inhibition of class A β-lactamases. This approach presents a potential solution to address antimicrobial resistance by allowing the design of inhibitors for specific classes of bacteria. However, the manuscript lacks sufficient details and explanations, raising concerns about the following points:

1.The choice of 30-mer peptides as inhibitors is not clarified. It is important to explain why this specific length was selected over shorter or longer peptide sequences and what advantages it offers.

The minimal length of the designed peptide must be greater than 1/f where f is characteristic frequency of the analysed biological function. This length is necessary to encompass at least one wavelength of the characteristic frequency. In case of beta-lactamases, characteristic frequency is found to be at f=0.0352 and thus the minimal length of designed peptides should be 1/0.0352=28.4. Thus, the length of 30 amino acids is just a little bit over the absolute minimum in length. Any longer sequence will have repetition of the same pattern and would have similar efficiency, but will be more expensive to be produced. The rationale to design 30-mer peptides is explained in “peptide design” section (page 13, lines 262-264). For clarity, we have underlined these lines.

2.The possible toxicity of the designed peptides is not addressed. It is essential to investigate and discuss any potential toxic effects that these inhibitors may have on human cells or beneficial bacteria.

This manuscript is a proof of concept to show that Resonance Recognition Model can be used to design beta-lactamase inhibitors. Now that our results have shown the functionality of this model, studying any potential toxic effects that pep3 may have on human cells or beneficial bacteria is the subject of ongoing further investigation in the laboratory and the results will be published elsewhere.

3.The manuscript does not explain how the peptides can avoid hydrolysis. Elaborating on methods that can be employed to prevent degradation will enhance the credibility of the proposed approach.

The peptides are designed to block beta-lactamase activity. Interaction between proteins (‘designed peptide’ and ‘beta-lactamase’) is based on resonant energy transfer between them at the specific frequency (characteristic RRM frequency f=0.0352). This has been explained in detail in our methodology section.

4.The sequences of the 30-mer peptides are not provided. It is crucial to include this information to allow other researchers to replicate and validate the findings.

The sequence details of the designed peptides pep1-4 are not publicly available due to the intellectual property process. On reasonable request, the corresponding author will make the sequence available.

5.The writing lacks sufficient explanations, which can make it difficult for readers to understand the methodology and results. Adding more detailed explanations, experimental procedures, and data interpretations will improve the clarity and impact of the manuscript.

For clarity, we have added more detailed explanations, experimental procedures, and data interpretations in the revised manuscript.

---

## [Editor Report · Decision Letter 1]

17 Aug 2023

Inhibition of β-lactamase function by de novo designed peptide

PONE-D-23-14582R1

Dear Dr. Mishra,

We’re pleased to inform you that your manuscript has been judged scientifically suitable for publication and will be formally accepted for publication once it meets all outstanding technical requirements.

Kind regards,

Farah Al-Marzooq, MD, PhD

Academic Editor

PLOS ONE

---

## [Editor Report · Acceptance letter]

29 Aug 2023

PONE-D-23-14582R1 

Inhibition of β-lactamase function by *de novo* designed peptide 

Dear Dr. Mishra:

I'm pleased to inform you that your manuscript has been deemed suitable for publication in PLOS ONE. Congratulations! Your manuscript is now with our production department. 

Kind regards, 

on behalf of

Dr. Farah Al-Marzooq 

Academic Editor

PLOS ONE